# A conceptual model on caregivers' hesitancy of topical fluoride for their children

Donald L. Chi[1,2]*, Darragh Kerr[1], Daisy Patiño Nguyen[1], Mary Ellen Shands[1], Stephanie Cruz[1], Todd Edwards[2], Adam Carle[3,4,5], Richard Carpiano[6], Frances Lewis[7]

1 Department of Oral Health Sciences, University of Washington School of Dentistry, Seattle, Washington, United States of America, 2 Department of Health Systems and Population Health, University of Washington School of Public Health, Seattle, Washington, United States of America, 3 James M. Anderson Center for Health Systems Excellence, Cincinnati Children's Hospital Medical Center, Cincinnati, OH, United States of America, 4 Department of Pediatrics, University of Cincinnati College of Medicine, Cincinnati, OH, United States of America, 5 Department of Psychology, University of Cincinnati College of Arts and Sciences, Cincinnati, Ohio, United States of America, 6 University of California Riverside, School of Public Policy, Riverside, California, United States of America, 7 Department of Child, Family and Population Health Nursing, University of Washington School of Nursing, Seattle, Washington, United States of America

* dchi@uw.edu

## Abstract

### Background

Topical fluoride hesitancy is a well-documented and growing public health problem. Despite extensive evidence that topical fluoride is safe and prevents tooth decay, an increasing number of caregivers are hesitant about their children receiving topical fluoride, leading to challenges in clinical settings where caregivers refuse preventive care.

### Purpose

To explore the determinants of topical fluoride hesitancy for caregivers with dependent children.

### Methods

In this qualitative study, we interviewed 56 fluoride-hesitant caregivers to develop an inductive conceptual model of reasons why caregivers are hesitant.

### Results

The core construct of the conceptual model of topical fluoride hesitancy centered on caregivers "wanting to protect and not mess up their child". Six domains comprised this core construct: thinking topical fluoride is unnecessary, wanting to keep chemicals out of my child's body, thinking fluoride is harmful, thinking there is too much uncertainty about fluoride, feeling pressured to get topical fluoride, and feeling fluoride should be a choice.

### Conclusions

Topical fluoride hesitancy is complex and multifactorial. Study findings provide insight for future efforts to understand and optimize caregivers' preventive care decision making.

**Data Availability Statement:** We are not able to share the qualitative interview scripts because the data contain potentially identifying or sensitive patient information. Data are available by contacting the University of Washington School of

Dentistry Office of Research (dentres@uw.edu) for researchers who meet the criteria for access to confidential data.

**Funding:** This study was funded in part by the U.S. National Institute of Dental and Craniofacial Research (NIDCR) grant nos. R01DE026741 and T90DE021984, the William T. Grant Foundation Scholars Program, and the Center for Advanced Study in the Behavioral Sciences (CASBS) at Stanford University. The funders had no role in study design, data collection and analysis, decision to publish, or preparation of the manuscript.

**Competing interests:** The authors have declared that no competing interests exist.

# Introduction

The U.S. Preventive Services Task Force recommends topical fluoride for young children and all state Medicaid programs pay for child topical fluoride treatments [1]. Although fluoride is safe and helps prevent tooth decay in children [2–8] a growing number of caregivers are hesitant about it, which has led to increasing rates of refusal during children's dental and medical visits [9,10]. Similar phenomenon exist in medicine with childhood vaccines and vitamin K [11–13] as well as with COVID-19 vaccines [14,15]. Drawing on the vaccination hesitancy literature [16], topical fluoride hesitancy is defined as *a delay in acceptance, thoughts of refusal, or refusal of topical fluoride despite availability*.

There are various explanations for fluoride hesitancy. Extrapolating from the COVID-19 vaccine hesitancy literature, low perceived need, concerns about its efficacy, and fear of adverse side effects may be common reasons [17,18]. At least one study identified differential prioritization of risks versus benefits as a reason for hesitancy [19]. Some caregivers may be skeptical about health providers who recommend topical fluoride without explaining why, which can lead to strained patient-provider communications, loss of trust, and reactance [20–24]. Recent studies indicate that reactance is also associated with resistance to mask use during COVID-19 [25,26]. Furthermore, beliefs about such preventive measures may be shared with other caregivers through social networks [27,28], which contributes to mis- and disinformation [23,29,30].

Science skepticism is a broader movement that has gained momentum in the U.S. and globally and contributes to the decision-making milieu of caregivers [31]. Some of this skepticism is related to documented scientific misconduct, including the retraction of a high-profile publication linking vaccines to childhood autism [32] which some interpret as an attempt to silence alternative views. There is also increasing scrutiny of overtreatment by providers seeking to increase revenue [33]. From a public health perspective, the concern is that such skepticism, which manifests through behaviors like fluoride hesitancy and refusal, signals growing barriers to public acceptance of scientific knowledge and innovations as well as care that is unnecessary [34]. The growing prevalence of fluoride hesitancy has implications for scientists developing therapies and technologies aimed at improving health as well as clinicians seeking to improve uptake of evidence-based care.

Despite the growing prevalence of topical fluoride hesitancy, no study to date has taken an inductive approach to understand the reasons why caregivers are hesitant. An inductive approach is one in which inherent themes from raw data are identified without the restraint of a specific a priori theoretical model [35]. In this study, we conducted in-depth caregiver interviews to develop a comprehensive conceptual model of why caregivers are hesitant about fluoride, with an emphasis on topical fluoride varnish commonly provided during preventive dental and medical visits.

# Materials and methods

## Study design and population

We adopted a phenomenological approach to capture the lived experiences of participants and our paradigm was postpositivist. We conducted a single occasion descriptive study that engaged fluoride-hesitant caregivers of children under age 18 years. We identified most study participants through a retrospective review of billing and health records at the University of Washington School of Dentistry and Seattle Children's Hospital. We used dental billing codes to identify children who had a dental examination between August 2016 and September 2018 and received no accompanying fluoride treatment. Health records were abstracted to verify

that the child did not receive fluoride. Additional participants were recruited via social media (e.g., Facebook, Twitter), dental and naturopathic medicine practices, and snowball sampling.

Study participants were eligible if they spoke English, were ≥18 years old, were the parent or legal guardian of a child under age 18 years with whom they lived, and reported being hesitant about topical fluoride for their child. Prior to the start of each interview, we provided the following a lay definition of topical fluoride as "the sticky stuff also called tooth vitamins or fluoride varnish that is painted onto teeth during visits to the dentist or doctor". To screen for hesitancy, caregivers were asked "On a scale of 1 to 10, with 1 being not opposed at all and 10 being totally opposed, how opposed are you to topical fluoride for your child or any of your children?" Caregivers who responded with a two or greater were eligible and scheduled for a telephone interview. We excluded caregivers who declined topical fluoride exclusively for financial reasons. The University of Washington Institutional Review Board approved the study (no number).

## Study procedures

Oral consent was obtained from all participants. One of three trained staff members conducted an interview guided by a 30-item semi-structured script that included questions on attitudes, beliefs, and knowledge about topical fluoride, beliefs about their child's oral health, and demographic information (S1 Table). The script was developed based on previous literature [9,36], pilot tested with three caregivers who were neutral to topical fluoride and were not subsequently interviewed as study participants, revised to improve clarity and flow, and finalized. Each study interview lasted between 20 and 90 minutes and was digitally recorded. Participants received a U.S. $30 gift card for participating.

## Measures

Topical fluoride hesitancy was operationalized with the question described above. We also included the survey item "Please tell me which of the following responses best describes you" (I absolutely want no topical fluoride for my child; most of the time I say no to topical fluoride for my child; sometimes I say no to topical fluoride for my child; I say yes to topical fluoride for my child, but I have thought about saying no).

## Data management and analyses

All interview data were transcribed verbatim by trained research assistants and verified for accuracy by a second research assistant. We used interpretivist grounded theory as the framework for analyzing the interview data [37]. Data were coded using a multi-phase hybrid inductive-deductive approach, which involves analyzing the data a priori without a guiding conceptual model at first and then developing and refining a model through multiple passes of the interview data [38]. We used content analytic techniques to inductively code participant responses to the main interview question about reasons for fluoride hesitancy, coding for the manifest not latent meanings of the words [39]. Two trained coders unitized the data. Open coding was used to group units into categories in which the unit shared a common property [37]. This was an iterative process in which the fit of the unit within and between categories was continuously assessed through constant comparative analysis to protect the independence of each category. Categories were labeled using participants' words (emics), defined, and then organized into broader domains. Domain labels were also given emic labels. Category and domain names and definitions were compiled into a codebook, which was then used to deductively code the responses to the remaining interview questions. Coders unitized the remaining data and coded the units into the established categories. A third member of the research team

reviewed and resolved coding discrepancies. To identify the core construct, team members reviewed all domains and categories to identify the concepts underlying topical fluoride hesitancy. The core construct was a given an emic name, defined, assessed for comprehensibility, and finalized.

## Reflexivity and trustworthiness

The three interviewers were trained researchers and not dentists, the latter of which helped to preserve a judgment-free environment in which caregivers could talk freely about their experiences with dentists. This was important given the reluctance many caregivers have in talking to dentists about sensitive topics related to fluoride decision making. To protect the trustworthiness of study results, we carried out three processes: 1) systematic in vivo multiple peer debriefing of categories, domains, and the core construct; 2) coding to consensus; and 3) maintaining an audit trail [40].

## Results and discussion

### Participant characteristics

We interviewed 56 caregivers. The mean age was 41.9±9.7 years (range: 29–79 years) (Table 1). Most caregivers were women (91.1%), white (57.1%), and non-Hispanic (87.5%). The mean number of children was 1.8±0.9 (range: 1 to 4) and the mean age of children was 8.0±4.3 years. Most caregivers had completed high school and 12.5% of caregivers reported an annual household income of < $25,000. Nearly one-third (29.3%) of caregivers with school-aged children reported their child as being eligible for free or reduced cost school meals. On a scale of 1 (not at all opposed) to 10 (totally opposed), the mean hesitancy score was 7.1±2.3 (range: 2 to 10). One-in-three caregivers with more than one child reported making decisions about topical fluoride differently for each child.

### Core construct

The core construct underlying caregiver topical fluoride hesitancy was *Wanting to Protect and Not Mess Up My Child*. The primary motivation for hesitancy caregivers expressed was a desire to protect their child from harm. Being the best caregiver possible meant that the caregiver balanced their fears and anxieties about refusing topical fluoride with the possibility that topical fluoride might benefit their child's oral health. As gatekeepers of their child's health, caregivers leaned toward conservative decision making, which in the extreme would manifest as refusing all forms of fluoride, including topical fluoride during dental or medical visits as well as fluoride from other sources like toothpaste and water. In the presence of alternative ways of preventing tooth decay, like limiting sugar intake or keeping teeth clean, caregivers viewed topical fluoride as unnecessary. In the presence of uncertainty on whether fluoride actually prevented tooth decay, topical fluoride was viewed as risky. In the presence of potential harm, topical fluoride was viewed as something to be avoided. Reinforcing this hesitancy was an underlying distrust in dentists, Western medicine, institutions, government, and the pharmaceutical industry, all of which were thought to be motivated primarily by self-interest and a potential threat to autonomous decision making.

### Conceptual model

The model consisted of six domains, each of which provide insight on why caregivers were hesitant about fluoride (Fig 1). A description of the domains and categories follow, including illustrative quotes from caregivers (Table 2).

**Table 1. Demographic characteristics of caregivers reporting topical fluoride hesitancy for their child or children (N = 56).**

| Variable | % (N) or Mean ± SD |
|---|---|
| Caregiver age (years) | |
| $\leq$ 35 | 28.6 |
| 36–50 | 55.4 |
| $\geq$ 51 | 16.1 |
| Caregiver gender | |
| Man | 8.9 |
| Woman | 91.1 |
| Caregiver self-reported race | |
| American Indian or Alaska Native | 3.6 |
| Asian | 10.7 |
| Black or African American | 8.9 |
| White | 57.1 |
| More than one race | 12.5 |
| Unreported | 7.1 |
| Caregiver self-reported ethnicity | |
| Hispanic | 10.7 |
| Non-Hispanic | 87.5 |
| Unreported | 1.8 |
| Number of children in household[a] | 1.8 ± 0.9 |
| Child age (years) | 8.0 ± 4.3 |
| Caregiver highest education level | |
| $\leq$ High school | 5.4 |
| Some college or 2-year degree | 32.1 |
| 4-year college degree | 37.5 |
| More than 4-year college degree | 25.0 |
| Annual household income | |
| < $25,000 | 12.5 |
| $25,000 to $75,000 | 35.7 |
| $75,000 or more | 37.4 |
| Child eligible for reduced-cost meals at school[b] | |
| Yes | 29.3 |
| No | 70.7 |

[a] Excluding children over 18 years old.

[b] Among households with at least one school-aged child.

## Domain 1: Thinking topical fluoride is unnecessary

Caregivers thought topical fluoride was unnecessary to keep their child's teeth healthy, especially if he or she had no cavities. Domain 1 consisted of five categories.

*1a. Thinking my child's teeth are fine without topical fluoride.* Caregivers did not think topical fluoride was needed if their child had few or no cavities. A 34-year-old caregiver (non-Hispanic white, score:4) stated, *"I don't really see the purpose of getting [topical] fluoride if they [my children] don't have cavities."* A small number of caregivers thought topical fluoride was unnecessary for baby teeth because these would eventually shed. Some caregivers regularly re-evaluated their child's need for topical fluoride. Although most preferred to avoid topical fluoride, some caregivers allowed it when needed. A 34-year-old caregiver (non-Hispanic white,

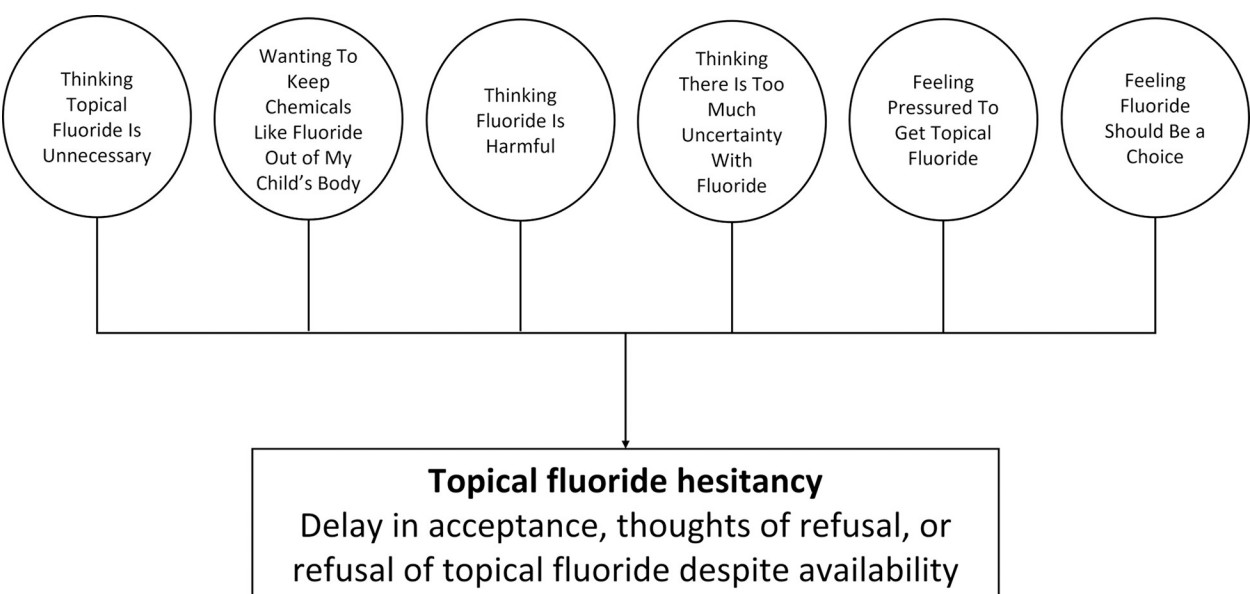

**Fig 1. Conceptual model on caregivers' hesitancy of topical fluoride for their children with six domains.**

score:7) said, *"If he [my son] has had. . .two consecutive cavity-free appointments, I would feel comfortable declining [topical fluoride]."*

**1b. Thinking topical fluoride is not effective.** Caregivers' confidence in the ability of topical fluoride to prevent cavities varied. Many caregivers attributed their child's cavities to genetics or to poor diet an oral hygiene routines rather than not getting topical fluoride. Some believed topical fluoride could not be expected to completely prevent cavities. A 41-year-old caregiver (non-Hispanic multiracial, score:5) said, "*There's [sic] just so many variables in life. It's hard to think one thing [topical fluoride] is going to solve everything.*" Others, however, believed topical fluoride was not at all effective. A 44-year-old caregiver (non-Hispanic white, score:6) shared, "*[My child] just continued to have bad teeth. . .I didn't feel like it [topical fluoride] was giving positive results to continue using it because it didn't seem to be making a difference.*"

**1c. Keeping your teeth clean is enough.** Caregivers discussed how good oral hygiene was sufficient in keeping their child's teeth healthy. This included making sure their child brushed, flossed, and had regular professional dental cleanings. As a 46-year-old caregiver (non-Hispanic American Indian/Alaska Native, score:10) summarized, "*I think it [preventing cavities] has a lot to do with keeping plaque off your teeth, not [putting] fluoride on them.*" Caregivers felt more comfortable declining topical fluoride if their child had good oral hygiene.

**1d. Having a healthy diet is more important than topical fluoride.** Caregivers also attributed good oral health to eating healthy foods rather than topical fluoride. A healthy diet was considered a safer, more predictable way to prevent cavities. For example, a 37-year-old caregiver (non-Hispanic Black, score:10) shared, "*If I find out she [my daughter] has a cavity or something of the sort, then we're going to cut down on whatever she's eating and indulging in. . .no fluoride [is] necessary.*" Many caregivers spoke about limiting sugar in their child's diet to prevent cavities.

**1e. Getting fluoride from other sources is enough.** Caregivers believed topical fluoride was unnecessary for their child because they were getting enough fluoride from other sources, such as toothpaste and water. These other sources of fluoride were considered sufficient to keep their child's teeth healthy. As a 39-year-old caregiver (non-Hispanic Asian, score:6) explained,

**Table 2. Domains and categories for a conceptual model of caregivers' hesitancy of topical fluoride for their children.**

| Model Domains | Domain Categories |
|---|---|
| **1. Thinking topical fluoride is unnecessary** | 1a. Thinking my child's teeth are fine without topical fluoride |
| | 1b. Thinking topical fluoride is not effective |
| | 1c. Keeping your teeth clean is enough |
| | 1d. Having a healthy diet is more important than topical fluoride |
| | 1e. Getting fluoride from other sources is enough |
| **2. Wanting to keep chemicals like fluoride out of my child's body** | 2a. Being careful about what goes into my child's body |
| | 2b. Worrying about my child ingesting topical fluoride |
| | 2c. Not wanting my child to have too much fluoride |
| **3. Thinking fluoride is harmful** | 3a. Believing fluoride is dangerous for my child's health |
| | 3b. Believing fluoride will damage the body |
| | 3c. Fearing fluoride will affect my child's developing mind |
| | 3d. Worrying fluoride will upset my child |
| **4. Thinking there is too much uncertainty about fluoride** | 4a. Feeling like I don't know enough about fluoride |
| | 4b. Hearing negative or conflicting things about fluoride |
| | 4c. Worrying fluoride has unknown effects |
| | 4d. Erring on the side of caution |
| **5. Feeling pressured to get topical fluoride** | 5a. Not getting the whole truth about topical fluoride |
| | 5b. Getting topical fluoride pushed on me |
| | 5c. Feeling like topical fluoride comes with an agenda |
| **6. Feeling fluoride should be a choice** | 6a. Having the right to decide what is best for my child |
| | 6b. Considering my child's opinion about topical fluoride |

*"I think they're [my children] getting fluoride in other places as well so I don't know if it's [topical fluoride] a must."* Caregivers reported it being easier to refuse topical fluoride than avoiding fluoridated water and toothpaste.

## Domain 2: Wanting to keep chemicals like fluoride out of my child's body

Caregivers wanted to keep their child's body free from chemicals like fluoride. Domain 2 consisted of three categories.

*2a. Being careful about what goes into my child's body.* Caregivers were cautious about exposing their child to fluoride. A 43-year-old caregiver (non-Hispanic white, score: 5) said *"I am often weary of doing things with chemicals [topical fluoride]."* Rather than making specific decisions about each substance their child could be exposed to, caregivers categorized them as synthetic, artificial, manmade, or unnatural–on the one hand–versus natural. Many caregivers desired a natural form of fluoride or a natural alternative to prevent cavities.

*2b. Worrying about my child ingesting topical fluoride.* Caregivers were concerned about their child swallowing topical fluoride. Based on guidance to avoid ingesting fluoride toothpaste, caregivers believed that topical fluoride should not be ingested either. Many believed topical fluoride would get absorbed through their child's mouth, accumulating throughout the

body over time. A 45-year-old caregiver (non-Hispanic biracial, score:10) explained, *"I don't want it [topical fluoride] going into my son's mouth because there's no way to keep that from possibly getting into his. . .system and build[ing] up."* Another 41-year-old caregiver (non-Hispanic white, score: 8) reasoned that, *"Our bodies aren't made to digest the chemical compounds that are in fluoride. . .[which is] just a waste product in our bodies that can end up in areas in the body that we can't fully digest and get rid of."*

**2c. Not wanting my child to have too much fluoride.** Caregivers also worried about their child getting too much fluoride. Rather than describing fluoride as inherently bad, caregivers espousing this concern thought that in general *"too much of a good thing is not necessarily a good thing."* They were uneasy about not knowing how much fluoride their child was getting compared to how much fluoride their child should get. In the absence of an objective recommended amount of fluoride, caregivers believed they needed to set the amount of fluoride to which their child was exposed, which varied greatly among caregivers. Some caregivers limited how often their child received topical fluoride or allowed some but not all sources of fluoride, and others eliminated all possible exposure. For example, a 34-year-old caregiver (non-Hispanic white, score:5) said, *"If we're living in a place where there is fluoride in the water. . .I'd be more hesitant to also. . .have it [topical fluoride] done at the dentist office."*

## Domain 3: Thinking fluoride is harmful

Caregivers were concerned about the negative consequences of fluoride, thinking it could harm their child physically, cognitively, and/or emotionally. Domain 3 consisted of four categories.

**3a. Believing fluoride is dangerous for my child's health.** Some caregivers described fluoride as generally *"not good for you"* and *"not a healthy option,"* but did not articulate specific side effects. Some described fluoride as a poison to the body, with a 35-year-old caregiver (non-Hispanic multiracial, score:5) stating, *"I think that topical fluoride can be toxic if given [at] high levels."* Caregivers were also concerned about their child *"becoming ill over time."*

**3b. Believing fluoride will damage the body.** Some caregivers were concerned about specific negative effects. Several caregivers worried that fluoride could cause irreversible damage such as *"chang[ing] our DNA."* As one 51-year-old caregiver (non-Hispanic white, score:7) said, *"fluoride is definitely, potentially carcinogenic."* Other caregivers believed it could damage specific body systems. A 34-year-old caregiver (Hispanic undisclosed race, score:10) shared, *"I just think that there is a chance that it [fluoride] could affect their [my child's] hormones."* Damage to their child's teeth, particularly dental fluorosis, was also a concern. One 37-year-old caregiver (non-Hispanic white, score:5) mentioned, *"I worry a little bit about if they [my children] get too much [fluoride] that they'll have permanent pitting or discoloration in their teeth."*

**3c. Fearing fluoride will affect my child's developing mind.** Many caregivers feared fluoride could affect their child's psychological, cognitive, and/or emotional development. One 39-year-old caregiver (non-Hispanic white, score:10) shared, *"[Fluoride] is a neurotoxin, which is known to lower IQ in children."* Others were concerned that fluoride would disrupt emotional development. A 32-year-old caregiver (Hispanic multiracial, score:5) stated, *"At a very pivotal age of someone [my son] who is emotionally developing and whose brain is literally growing, things like fluoride do cause me concern."* Caregivers of children with special healthcare needs worried that fluoride exposure could make their child's condition worse. A 34-year-old caregiver (Hispanic undisclosed race, score:10) shared, *"[My son is] already significantly delayed in his [brain] development. I can't risk a chance that it [his brain development] could be hindered any further even if it's just a little bit [of fluoride]."*

*3d. Worrying topical fluoride will upset my child.* Caregivers described their child's negative experiences with topical fluoride and wanting to avoid future bad experiences. This was a particular concern for caregivers of children with special healthcare needs. As a 42-year-old caregiver (non-Hispanic white, score:7), whose child has autism, mentioned, *"I wouldn't want to put him through anything else [at the dentist] that would be even harder to endure. . .or uncomfortable."* Another 35-year-old caregiver (non-Hispanic white, score:6) explained, *"My son is on the autism spectrum, and he is very sensitive.. . .[getting topical fluoride is] such an intense pain [for him].* A 31-year-old caregiver (non-Hispanic white, score:5) shared, "*they [dental staff] offer fluoride at the end of a visit, and. . .at this point, [my son] was upset and crying, so I just said no because [my son] was emotional.*"

## Domain 4: Thinking there is too much uncertainty about fluoride

Many caregivers felt uncertainty about fluoride. Domain 4 consisted of four categories.

*4a. Feeling like I don't know enough about fluoride.* Caregivers worried they did not know enough about the purpose, benefits, and possible side effects of fluoride, which made them uncertain about its safety and effectiveness. A 41-year-old caregiver (non-Hispanic multiracial, score:3) shared, *"I don't know anything about it [topical fluoride]. . .that's why it scares me."* Other caregivers felt uncomfortable that their child's dentist recommended it but did not explain why. As one 29-year-old caregiver (non-Hispanic Black, score:10) said, *"I don't think there's enough education. They [dentists] don't really explain it [topical fluoride]."*

*4b. Hearing negative or conflicting things about fluoride.* Caregivers explained that hearing negative or conflicting information about fluoride made them feel uncertain. When researching information about fluoride safety, a 61-year-old caregiver (non-Hispanic white, score:10) described finding *". . .just as much information that was against fluoride as I found stuff that was for it."* This prompted some caregivers to be cautious about fluoride-related decisions. Caregivers considered advice from trusted health professionals and family. As one 41-year-old caregiver (Hispanic biracial, score:7) shared, *"I had a couple of [medical] doctors. . .who alerted me to when she [my daughter] was small, saying you shouldn't let her get fluoride in her teeth."*

*4c. Worrying fluoride has unknown effects.* Caregivers worried about the unknown effects of fluoride and how these unknowns might impact their child in the future. They spoke about there not being enough evidence that fluoride would not have long-term health consequences. They were not *". . .confident about the studies that have and have not been done about it [topical fluoride] and its effects."* Caregivers of young children were particularly worried, with one 52-year-old caregiver (non-Hispanic white, score:5) sharing, "*I don't believe that fluoridated varnish is right to put it inside a child's system, which is growing. . .we don't know the future of that."* A 40-year-old caregiver (non-Hispanic white, score:8) said, *"Because of the unknown lifetime effects [of topical fluoride], we [my partner and I] don't want to take a risk of something [like topical fluoride]."* The uncertainty of fluoride potentially harming their child prompted a 41-year-old caregiver (non-Hispanic multiracial, score:3) to share that *"It's difficult to know until later what's really happening. . .you just try your best and pray you don't mess up your kids."*

*4d. Erring on the side of caution.* When making decisions about topical fluoride, caregivers weighed risks and benefits. Although many caregivers acknowledged the oral health benefits of topical fluoride, the uncertainty around its safety was too great a risk for some. For caregivers who were uncertain, it was better to avoid topical fluoride than to accept both the potential risks and benefits. As one 61-year-old caregiver (non-Hispanic white, score:10) explained, *"I'd rather err on the side of things I know about like [avoiding] sticky candies. . .than [accepting]*

*something that I didn't quite understand."* For one 41-year-old caregiver (Hispanic biracial, score:7), *"The benefits for stronger teeth don't. . .equal the high cost[s]."* Another 38-year-old caregiver (Hispanic undisclosed race, score:9) stated it more bluntly, *"I'd rather him [my son] get a cavity than cancer [from fluoride]."*

## Domain 5: Feeling pressured to get topical fluoride

Caregivers spoke about feeling pressured to accept topical fluoride, which prompted many to question its necessity and the motives of those who recommended it. Domain 5 consisted of three categories.

*5a. Not getting the whole truth about topical fluoride.* Some caregivers were hesitant about topical fluoride because they felt that dentists were biased and untrustworthy. They believed dentists only presented the benefits of topical fluoride and not the risks, which made caregivers suspicious of dentists' motives. As a 37-year-old caregiver (non-Hispanic Black, score:10) shared, *"When I ask my doctor about fluoride, there's always something [information] that [my doctor] tells me about the good but they never have anything that touches about the stuff that's not really good, so it sort of makes me uneasy. . .I feel like it's deception."* Several caregivers did not believe dentists kept current with research on fluoride making them skeptical of dentists' expertise on the subject. A 43-year-old caregiver (non-Hispanic white, score:5) said, *"My experience with lots of doctors and dentists is that they sometimes just use what they learned when they were in school and they never. . .update their practices when new information is coming out."*

*5b. Getting topical fluoride pushed on me.* Caregivers felt that dentists presented topical fluoride as non-optional, which contributed to hesitancy. A 30-year-old caregiver (non-Hispanic biracial, score:10) said, *"I feel like they [dentists] want you to say okay to [topical fluoride] and get it regardless of any reservations that I have."* Caregivers spoke about dentists applying topical fluoride on their child's teeth without asking. One 43-year-old caregiver (non-Hispanic white, score: 5) recalled their child's dentist *"just goes and does it [applies topical fluoride] and doesn't really ask for my permission."* Others felt coerced into accepting it, as a 61-year-old caregiver (non-Hispanic white, score:10) described, *"He [the dentist] refused to give my child even a tooth cleaning unless I agreed to fluoride."*

*5c. Feeling like topical fluoride comes with an agenda.* Caregivers felt dentists, the pharmaceutical industry, and the government had motives to promote fluoride. They believed financial gain was the main motivator rather than the promotion of good oral health. One 49-year-old caregiver (non-Hispanic Asian, score:8) said, *"The dentists are touting about how great it [topical fluoride] is but it's just something extra for them to charge for."* Some caregivers identifying as Black or American Indian mentioned past injustices and persisting systemic racism that led them to distrust the government and other Western institutions. A 46-year-old caregiver (non-Hispanic American Indian/Alaska Native, score:10) explained, *"I feel like Western medicine is lying to our people [Indigenous people] about what fluoride does to our people. . .[It] poisons communities of color."*

## Domain 6: Feeling fluoride should be a choice

Caregivers spoke about feeling like fluoride should be their choice and that their child should also have a voice in the decision-making process. Domain 6 consisted of two categories.

*6a. Having the right to decide what is best for my child.* Caregivers stressed the importance of having autonomy over health care decisions affecting their child. They viewed all health care for their child as optional, including topical fluoride. As a 34-year-old caregiver (non-

Hispanic white, score:4) said, *"I have the choice of a health care decision for my child, I don't really need to do anything."*

*6b. Considering my child's opinion about topical fluoride.* When making decisions about topical fluoride, some caregivers talked about considering their child's preferences, underscoring the importance of reinforcing the child's autonomy in health care decisions. A 49-year-old caregiver (non-Hispanic Asian, score:8) said, *"He [my son] didn't want it [topical fluoride], so I complied."* Some caregivers noted that because their child was getting older, they could independently make decisions about their oral health. As one 37-year-old caregiver (non-Hispanic Black, score:10) mentioned, *"My oldest daughter, she's of that age where she can make appointments and things for herself. So if. . .the dentist asks her if she wants to use fluoride, my daughter can make that opinion for herself and use fluoride."* Most of these caregivers, however, listened to their child's opinion about topical fluoride because their child *"just really didn't like that sticky feeling in [their] mouth"* or said *"it was really gross tasting."*

Our goal was to develop a conceptual model on reasons caregivers are hesitant. The main finding is that topical fluoride hesitancy is a complex multifactorial behavior motivated by a caring and protective caregiver. The study implications are that dentists' views of fluoride-hesitant caregivers need to shift. Our model provides potential targets to improve caregiver-provider communications to ensure that caregivers are better positioned to make fluoride-related decisions for their child.

Participating caregivers demonstrated a nuanced skepticism about fluoride and typically subscribed to multiple model domains. This profile runs counter to typical provider perceptions [36]. Many dentists do not ask caregivers why they are hesitant and instead assume that a hesitant caregiver simply lacks knowledge about fluoride [36]. Providing education alone is necessary but insufficient in changing patient health behaviors [41]. Such an approach might be interpreted by some caregivers as being insensitive. Mis-specified education could also come across as dismissive or make hesitant caregivers feel like they are being pressured to accept fluoride, leading to reactance. Dentists and other health providers who recommend topical fluoride should acknowledge the complexity of hesitancy behaviors by relinquishing any assumptions about hesitant caregivers, asking open-ended questions to understand why the caregiver is hesitant, and responding to the caregiver's specific concerns [36]. Additional research is needed on ways in which dentists should frame communications and behavioral approaches with hesitant caregiver to effectively address underlying concerns expressed by caregivers while preserving decision-making autonomy [42].

The core construct of our conceptual model centered on caregivers wanting to protect their children. Health providers may perceive fluoride-hesitant caregivers as being susceptible to conspiracy theories, and by extension, as low literacy, illogical, and uncaring [43]. While conspiracy theory type views were present in our interview data they were uncommon. Similarly, past work indicates that dentists perceive non-traditional "Hippie" and "granola" parenting styles to be associated with caregiver refusal of topical fluoride [10]. Thus, it is reasonable to assume that many dentists may not take fluoride-hesitant caregivers' concerns seriously. It would not be a stretch to surmise that granola caregivers are viewed as either detached from one's parenting responsibilities or unreasonably and intensively involved with decisions that affect their child [44]. Either way, fluoride-hesitant caregivers are likely to be judged negatively, especially for thinking about or making health care decisions that could harm their child. Our core concept underscores the importance of all providers starting with the premise that caregivers are making informed health decisions in the best interest of their child and that hesitancy is not an indication of a recalcitrant or resistant caregiver [45]. Future work should seek to understand and reduce the decision making-related biases that health providers carry that influence patient-provider communications and relations.

Study results suggest two interrelated areas in which dentists' practice behaviors may need to shift. The first area is that not all children need topical fluoride, which is consistent with current American Academy of Pediatric Dentistry policies that recommend topical fluoride for children at risk for caries [46], unlike vaccines for which a population-level benefit is derived from achieving herd immunity. This is consistent with caregiver concerns that topical fluoride may be unnecessary (Domain 1). Contrary to current practice, in which most–if not all–children are offered topical fluoride during preventive dental visits, the decision to recommend fluoride should be judicious and risk based. Such an approach would be consistent with evidence-based guidelines and address caregiver concerns about the need for topical fluoride during dental visits, especially for low-risk children (e.g., those who have never had a cavity, healthy behaviors), and reduce costs for families, payors, and federal insurance programs like Medicaid [46].

The second area is that dentists should improve the way that risk for cavities is described to caregivers. Risk is a function of past and current disease, independent risk factors (e.g., dietary sugar, fluoride exposure), and the interplay between risk factors. There is no silver bullet that prevents cavities and topical fluoride is not a panacea, especially for children with high added sugar intake, which may be related to caregivers' misunderstanding about the effectiveness of topical fluoride (Domain 4). Rather, topical fluoride is part of a multi-pronged strategy to prevent tooth decay and comes in forms other than the type provided by dentists, including water and toothpaste. A persisting challenge is that many caregivers may have difficulty altering a child's dietary behaviors [47]. Additional work is needed on how to restructure delivery of appropriate risk-based care and risk communication strategies.

Future behavioral and social science research aimed at understanding phenomenon like fluoride hesitancy should continue to adopt a multidisciplinary approach, which would help to document the various levels at which hesitancy can operate (e.g., social, community, individual). The intended outcome for interventions aimed at hesitant caregivers is not necessarily acceptance of treatments like fluoride. Rather, the goal is to define appropriate communication strategies that align with specific concerns underlying hesitancy that health providers can use to facilitate meaningful dialogue with caregivers. Open dialogue is a starting point. Dentists should provide interested caregivers with the information and resources needed for them to make fluoride decisions for their own child in a judgment-free context, particularly given the uncertainty caregivers may have about fluoride. There is also a need to understand how provider trust is established and eroded. A multidisciplinary approach is likely to be relevant in understanding the determinants of hesitancy caregivers may have about other types of preventive treatment like long-standing childhood vaccines and more recently developed COVID-19 vaccines.

There were three main study limitations. First, participants were from a convenience sample recruited mainly through two pediatric dentistry clinics in the same city. We conducted interviews until we reached saturation on themes. Second, our model is being applied to topical fluoride hesitancy, which is a broad concept that includes caregivers who may accept topical fluoride but have concerns as well as caregivers who refuse topical fluoride. This is an important distinction because interventions for mostly accepting but still hesitant caregivers might focus on addressing modifiable concerns whereas intervention for refusing caregivers would instead be aimed at offering acceptable alternative strategies in lieu of fluoride. Future work should continue to elucidate how fluoride hesitancy is related to refusal behaviors. Third, we did not have caries risk data to validate the extent to which caregiver hesitancy might be warranted. Additional studies are needed to examine how caries risk and presence of dental diseases are related to topical fluoride hesitancy and refusal.

## Conclusions

In conclusion, study findings indicate that topical fluoride hesitancy is a complex behavior with multiple inputs, all of which center on caregivers wanting to protect their child. Future research should continue to investigate how communication and behavioral strategies can be developed to address specific caregiver concerns, the ultimate goal of which is to equip caregivers with both knowledge and resources to make decisions that optimize child health outcomes.

## Supporting information

**S1 Table. Semi-structured interview guide used to collect data from caregivers reporting topical fluoride hesitancy for their child or children.**
(PDF)

**S2 Table. Domains, categories, and corresponding definitions for a conceptual model of topical fluoride hesitancy.**
(PDF)

## Acknowledgments

We would like to thank all the participating caregivers for sharing their time and expertise.

## Author Contributions

**Conceptualization:** Donald L. Chi, Frances Lewis.

**Data curation:** Stephanie Cruz.

**Formal analysis:** Donald L. Chi, Darragh Kerr, Daisy Patiño Nguyen, Mary Ellen Shands, Todd Edwards, Adam Carle, Richard Carpiano.

**Funding acquisition:** Donald L. Chi.

**Investigation:** Donald L. Chi, Darragh Kerr.

**Project administration:** Donald L. Chi.

**Resources:** Donald L. Chi.

**Supervision:** Donald L. Chi.

**Writing – original draft:** Donald L. Chi, Darragh Kerr.

**Writing – review & editing:** Donald L. Chi, Daisy Patiño Nguyen, Mary Ellen Shands, Stephanie Cruz, Todd Edwards, Adam Carle, Richard Carpiano, Frances Lewis.

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
