## [Decision Letter · Decision Letter 0]

6 Feb 2023

PONE-D-22-35556A conceptual model on caregivers’ hesitancy of topical fluoride for their childrenPLOS ONE

Dear Dr. Chi,

Thank you for submitting your manuscript to PLOS ONE. After careful consideration, we feel that it has merit but does not fully meet PLOS ONE’s publication criteria as it currently stands. Therefore, we invite you to submit a revised version of the manuscript that addresses the points raised during the review process.

We look forward to receiving your revised manuscript.

Kind regards,

Boyen Huang, DDS, MHA, PhD

Academic Editor

PLOS ONE

Journal Requirements:

"This study was funded in part by the U.S. National Institute of Dental and Craniofacial Research (NIDCR) grant nos. R01DE026741 and T90DE021984, the William T. Grant Foundation Scholars Program, and the Center for Advanced Study in the Behavioral Sciences (CASBS) at Stanford University."

"This study was funded in part by the U.S. National Institute of Dental and Craniofacial Research (NIDCR) grant nos. R01DE026741 and T90DE021984, the William T. Grant Foundation Scholars Program, and the Center for Advanced Study in the Behavioral Sciences (CASBS) at Stanford University."

"This study was funded in part by the U.S. National Institute of Dental and Craniofacial Research (NIDCR) grant nos. R01DE026741 and T90DE021984, the William T. Grant Foundation Scholars Program, and the Center for Advanced Study in the Behavioral Sciences (CASBS) at Stanford University."

**Additional Editor Comments:**

Dear Dr. Chi,

Thank you for submitting your manuscript to PLOS ONE. I enjoyed reading your work. Please see the reviewers' comments and respond accordingly.

Regards,

Boyen Huang

Reviewers' comments:

Reviewer's Responses to Questions

**Comments to the Author**

1. Is the manuscript technically sound, and do the data support the conclusions?

Reviewer #1: Yes

Reviewer #2: Yes

2. Has the statistical analysis been performed appropriately and rigorously? 

Reviewer #1: N/A

Reviewer #2: N/A

3. Have the authors made all data underlying the findings in their manuscript fully available?

Reviewer #1: Yes

Reviewer #2: No

4. Is the manuscript presented in an intelligible fashion and written in standard English?

Reviewer #1: Yes

Reviewer #2: Yes

5. Review Comments to the Author

Reviewer #1: Could not access Appendix A as referred to in the text. Received no response from Plos One when requested this. Definitions are required of methods used and references citations need to corrected. See other comments on attached feedback.

Reviewer #2: This is a well-written article on an important topic in clinical dentistry. The methods are sound and the data are presented well. It was an enjoyable read.

The authors state that some carers have conspiracy theory type views, whereas others’ hesitancy could really be consistent with clinical guidelines anyway. What will help the general reader is a clear statement of the current guidelines for topical fluoride application, presumably from Reference 46 or the American Dental Association or equivalent bodies. Furthermore, what is the current practice with fluoride application at the study site? Is topical fluoride recommended for all children? Do clinicians take steps that are necessary to prevent fluorosis?

The text in the two paragraphs of the Discussion on reshaping dental practitioner behavior (p22-23) cover important contents and should appear earlier on. A long explanation (discussion) carries the risk of such important messages being lost in the details. Some of the text on future studies could appear much later. A more succinct explanation could also help to keep readers interested.

An important area to explore would be mapping carer’s attitude with clinical data on caries risk and provider’s recommendation for topical fluoride application. I realise this is probably out of the scope of the current project, in which case the limitation should be acknowledged in the Discussion section.

I have made some more relatively minor suggestions below.

P3, last paragraph: The sentence ‘Patients have also increasingly scrutinized providers for promoting treatments that generate income [33]’ could be better expressed using the term ‘overtreatment’. It is not only patients but also the general public, academics and regulatory bodies that are scrutinizing overtreatment, so the sentence could be better written as, “There is also an increasing scrutiny over overtreatment ….”.

P5: [9, 35-36]

P17, Domain 5: … to question its necessity…

P19: Non-US readers may not understand what AIAN means, so please use its full form with abbreviation in brackets the first time it appears in text. Alternatively, if it does not appear elsewhere in the text, please just use the full form.

P21: Please simplify this sentence as it is a bit too long, “Such an approach might be interpreted by some caregivers as being insensitive, but mis-specified

education could also come across as dismissive or make hesitant caregivers feel like they are

being pressured to accept fluoride, leading to reactance.”

References: Please ensure formatting consistency, e.g., sentence case or first letter capitalization for title, and inclusion or exclusion of publication date.

Reference 41 – please insert web-link.

6. PLOS authors have the option to publish the peer review history of their article (what does this mean?). If published, this will include your full peer review and any attached files.

Reviewer #1: No

Reviewer #2: No

---

## [Author Response · Author response to Decision Letter 0]

10 Feb 2023

Reviewers' comments:

Reviewer's Responses to Questions

Comments to the Author

1. Is the manuscript technically sound, and do the data support the conclusions?

Reviewer #1: Yes

Reviewer #2: Yes

2. Has the statistical analysis been performed appropriately and rigorously? 

Reviewer #1: N/A

Reviewer #2: N/A

3. Have the authors made all data underlying the findings in their manuscript fully available?

Reviewer #1: Yes

Reviewer #2: No

We are not able to share the qualitative interview scripts because the data contain potentially identifying or sensitive patient information. We have provided institutional contact information to which data requests may be sent.

4. Is the manuscript presented in an intelligible fashion and written in standard English?

Reviewer #1: Yes

Reviewer #2: Yes

5. Review Comments to the Author

Reviewer #1: Could not access Appendix A as referred to in the text. Received no response from Plos One when requested this. Definitions are required of methods used and references citations need to corrected. See other comments on attached feedback.

We apologize for this oversight. Both Appendix A and B have been provided as Supplementary Table 1 and Supplementary Table 2. Revisions have been made to address concerns related to the Methods section and reference citations. We have also made revisions based on comments on the PDF. Thank you very much for the careful review.

Reviewer #2: This is a well-written article on an important topic in clinical dentistry. The methods are sound and the data are presented well. It was an enjoyable read.

The authors state that some carers have conspiracy theory type views, whereas others’ hesitancy could really be consistent with clinical guidelines anyway. What will help the general reader is a clear statement of the current guidelines for topical fluoride application, presumably from Reference 46 or the American Dental Association or equivalent bodies. Furthermore, what is the current practice with fluoride application at the study site? Is topical fluoride recommended for all children? Do clinicians take steps that are necessary to prevent fluorosis?

Thank you for this clarifying point. The relevant part of the Discussion section has been expanded to make it clear that the American Academy of Pediatric Dentistry recommends topical fluoride for children at risk for caries (and not all children). Language was included that almost all clinics offer topical fluoride to all children. Topical fluoride is not recommended for all children. Fluorosis is related to chronic ingestion of fluoride, and therefore the topical fluoride provided during medical and dental visits is not considered a risk factor.

The text in the two paragraphs of the Discussion on reshaping dental practitioner behavior (p22-23) cover important contents and should appear earlier on. A long explanation (discussion) carries the risk of such important messages being lost in the details. Some of the text on future studies could appear much later. A more succinct explanation could also help to keep readers interested.

We agree the discussion points about reshaping providers’ behaviors is important. However, we wanted to place these points after discussing the main findings related to the conceptual model, which was the primary aim of the study. Therefore, placement of these paragraphs was not altered. We agree parts of the discussion section could be cut to reduce wordiness and keep readers interested. These changes are reflected in the revised paper.

An important area to explore would be mapping carer’s attitude with clinical data on caries risk and provider’s recommendation for topical fluoride application. I realise this is probably out of the scope of the current project, in which case the limitation should be acknowledged in the Discussion section.

Unfortunately, we do not have clinical data on the children. But this is an excellent point and is the topic of future investigations. This is included as a study limitation.

I have made some more relatively minor suggestions below.

P3, last paragraph: The sentence ‘Patients have also increasingly scrutinized providers for promoting treatments that generate income [33]’ could be better expressed using the term ‘overtreatment’. It is not only patients but also the general public, academics and regulatory bodies that are scrutinizing overtreatment, so the sentence could be better written as, “There is also an increasing scrutiny over overtreatment ….”.

We agree. This has been fixed as suggested.

P5: [9, 35-36]

This has been fixed.

P17, Domain 5: … to question its necessity…

Done.

P19: Non-US readers may not understand what AIAN means, so please use its full form with abbreviation in brackets the first time it appears in text. Alternatively, if it does not appear elsewhere in the text, please just use the full form.

Great point. This has been revised.

P21: Please simplify this sentence as it is a bit too long, “Such an approach might be interpreted by some caregivers as being insensitive, but mis-specified

education could also come across as dismissive or make hesitant caregivers feel like they are

being pressured to accept fluoride, leading to reactance.”

This has been broken into 2 sentences.

References: Please ensure formatting consistency, e.g., sentence case or first letter capitalization for title, and inclusion or exclusion of publication date.

Done.

Reference 41 – please insert web-link.

Done.

6. PLOS authors have the option to publish the peer review history of their article (what does this mean?). If published, this will include your full peer review and any attached files.

Do you want your identity to be public for this peer review? For information about this choice, including consent withdrawal, please see our Privacy Policy.

Reviewer #1: No

Reviewer #2: No

---

## [Decision Letter · Decision Letter 1]

24 Feb 2023

A conceptual model on caregivers’ hesitancy of topical fluoride for their children

PONE-D-22-35556R1

Dear Dr. Chi,

We’re pleased to inform you that your manuscript has been judged scientifically suitable for publication and will be formally accepted for publication once it meets all outstanding technical requirements.

Kind regards,

Boyen Huang, DDS, MHA, PhD

Academic Editor

PLOS ONE

Additional Editor Comments (optional):

Reviewers' comments:

Reviewer's Responses to Questions

**Comments to the Author**

1. If the authors have adequately addressed your comments raised in a previous round of review and you feel that this manuscript is now acceptable for publication, you may indicate that here to bypass the “Comments to the Author” section, enter your conflict of interest statement in the “Confidential to Editor” section, and submit your "Accept" recommendation.

Reviewer #1: All comments have been addressed

Reviewer #2: All comments have been addressed

2. Is the manuscript technically sound, and do the data support the conclusions?

Reviewer #1: Yes

Reviewer #2: Yes

3. Has the statistical analysis been performed appropriately and rigorously? 

Reviewer #1: N/A

Reviewer #2: N/A

4. Have the authors made all data underlying the findings in their manuscript fully available?

Reviewer #1: Yes

Reviewer #2: Yes

5. Is the manuscript presented in an intelligible fashion and written in standard English?

Reviewer #1: Yes

Reviewer #2: Yes

6. Review Comments to the Author

Reviewer #1: Thank you to the authors for addressing the reviewers concerns. This article is now worthy of publication and will challenge clinicians to provide evidence based and holistic care rather than the seemly current approach to apply fluoride to all.

Reviewer #2: (No Response)

7. PLOS authors have the option to publish the peer review history of their article (what does this mean?). If published, this will include your full peer review and any attached files.

Reviewer #1: No

Reviewer #2: No

---

## [Editor Report · Acceptance letter]

28 Feb 2023

PONE-D-22-35556R1 

A conceptual model on caregivers’ hesitancy of topical fluoride for their children 

Dear Dr. Chi:

I'm pleased to inform you that your manuscript has been deemed suitable for publication in PLOS ONE. Congratulations! Your manuscript is now with our production department. 

Kind regards, 

on behalf of

Dr Boyen Huang 

Academic Editor

PLOS ONE